

# Surface Ozone Distribution & Trends Over Ireland : Insights from long-term measurement record and source attribution modelling

Nikhil Korhale [1], Tabish Ansari [2], Tim Butler [2], Jurgita Ovadnevaite[1], Colin D.O'Dowd[1] , Liz Coleman[1]

1 School of Natural Sciences, Physics, Ryan Institute's Centre for Climate & Air Pollution Studies, University of Galway, Galway, Ireland

2 Research Institute for Sustainability - Helmholtz Centre Potsdam, Potsdam, 14467, Germany

**Corresponding author :**

 Name -   Dr. Liz Coleman

 Email Id – liz.coleman@universityofgalway.ie

**Abstract**

Surface ozone ($O_3$) pollution is assessed across Ireland with a focus on long-term trends with a specific focus on the Mace Head atmospheric research station which monitors background $O_3$ advected into Europe via prevailing South Westerlies. Using innovative trajectory analysis, $O_3$ concentrations, exceedances and were identified by sectors, revealing distinct seasonal and spatial patterns. Findings show a significant rising trend in surface $O_3$ at Irish urban sites over the past two decades but without a similar trend at coastal sites. Highest $O_3$ levels and exceedances were observed at remote coastal sites, less influenced by local emissions, and heavily influenced by meteorological processes, including transboundary pollution and stratospheric intrusion. At Mace Head, springtime $O_3$ levels show a declining trend, with a



rising winter-time trend. Looking only at the clean sector, the springtime decline remains
significant; but without rising wintertime trends, implying the rising winter trends are a
response to declining European emissions. Advanced modelling tools are used to quantify $O_3$
source contributions, elucidating key drivers behind the observed changes. Characteristic
springtime $O_3$ maxima at Mace Head are attributed to stratospheric transport, influences from
westerly transboundary air pollution, and lightning $NO_x$. Combined trend and sectoral
observational analysis reveals that total spring-time concentrations are in decline, with
exceedances from the UK & continental sector declining at a greater rate. This research
highlights the importance of seasonal factors in air quality management across Ireland,
emphasising the need for a multi-faceted approach to control $O_3$ levels and reduce exceedances
through global and regional emission reductions.
**Keywords -** Meteorology, $NO_x$, Climate, $CH_4$, Emissions, VOC.
**1. Introduction**
Surface Ozone ($O_3$) has significant implications for health, vegetation, and climate. As $O_3$ is
highly reactive, its chemical production is driven by complex photochemical processes,
responding non-linearly to pollution control, creating challenges for its effective regulation.
Elevated $O_3$ levels cause severe health issues, prolonged exposure to high $O_3$ levels is linked
to respiratory issues, cardiovascular problems, and reduced lung function, particularly in
sensitive populations such as children, the elderly, and individuals with pre-existing respiratory
conditions (Lin et al., 2018; Todorović et al., 2019; Zhang et al., 2019, WHO, 2021). $O_3$
pollution can adversely impact vegetation by reducing agricultural productivity (Ashmore et
al., 2005; Paoletti et al., 2006). $O_3$ is also the third most significant greenhouse gas after Carbon
Dioxide ($CO_2$) and methane ($CH_4$), contributing to climate instability (IPCC, 2021). The
lifetime of $O_3$ in the free troposphere is on the order of several weeks, and it is affected by



large-scale atmospheric circulation patterns (Wespes et al., 2017), and meteorological factors
such as temperature, solar radiation, wind speed, and atmospheric stability play a significant
role in $O_3$ formation. (Ding et al., 2023; Khiem et al., 2010).
$O_3$ is formed in the atmosphere from precursors Nitrogen Oxides ($NO_x$), carbon monoxide
(CO), and volatile organic compounds (VOCs) through photochemical reactions. The reactive,
interdependent atmospheric chemistry leads to a non-linear relationship between $O_3$ and its
precursors (Seinfeld and Pandis, 2016), and effective $O_3$ mitigation requires an understanding
of processes influencing $O_3$ production and removal mechanisms (Fowler et al., 2013). $NO_x$
can suppress or enhance $O_3$ formation, depending on the atmospheric chemistry regime. In
polluted urban environments, high $NO_x$ emissions can lead to $O_3$ dissociation, retarding
formation whereas in relatively clean environments, $O_3$ formation is correlated with $NO_x$
concentration (Seinfeld and Pandis, 1997;Tavella & da Silva Júnior, 2021). Seasonal and
regional variations further complicate the regulation, with higher $O_3$ levels observed in summer
across the northern hemisphere due to increased temperatures, solar radiation, and abundant
precursors (Moiseenko et al., 2021; Sicard et al., 2016). In marine boundary layers, $O_3$ levels
are generally lower than in continental regions, though specific oceanic environments can
exhibit high $O_3$ concentrations due to inflows from polluted areas (Boylan et al., 2014; Girach
et al., 2020). Another factor which influences O3 level is the North Atlantic Oscillation (NAO
which influences O3 levels in Western Europe, with positive phases enhancing the transport of
O3 and precursors from North America. This effect is particularly notable in southwest, central,
and northern Europe  (Bonaccorso et al., 2015; Creilson et al., 2003; Pausata et al., 2012).
While Ireland's air quality is mostly governed by the influx of clean maritime air from the
Atlantic Ocean (Tripathi et al., 2010), particular synoptic scenarios allow for the intrusion of
polluted air masses from continental Europe. These events, though infrequent, can bring





substantial amounts of ozone and its precursors ($NO_x$, VOCs), contributing to short-term $O_3$
pollution episodes.
The World Health Organisation (WHO) publishes Air Quality Guidelines (AQGs) as a non-
legally binding global target for governments to achieve within their jurisdictions. These AQGs
comprise evidence-based recommendations of limit values to protect public health. The current
recommended AQGs for $O_3$ is expressed as a daily maximum of 8-hourly running average $O_3$
value of 100 µg/m³. Days when $O_3$ levels exceed the recommended AQGs are classified as
exceedance days. Factors contributing to exceedances include high solar radiation, stagnant air
masses, and local emissions and regional and transboundary transport of $O_3$ and precursors.
Over the past 150 years, there has been a 40% increase in $O_3$ levels owing to rising precursor
emissions. (Archibald et al., 2020; Griffiths et al., 2021; Young et al., 2013). Despite European
Union emission reduction policies, $O_3$ pollution remains a problem, with over 94% of those
living in European cities exposed to $O_3$ levels exceeding the WHO AQGs in 2022 (EEA 2024,
WHO 2021). Over 22,000 premature deaths in the EU were attributable to short-term exposure
to $O_3$ in 2021(Soares et al., 2023).
Long-term data from the Mace Head research station in Ireland reveal seasonal peaks in $O_3$
during spring and lows in summer. (Derwent, 1998; Derwent et al., 1994, 2018a). Historical
trends show increasing baseline $O_3$ levels in the 1980s and 1990s, stability in the 2000s, and a
decline in the 2010s.(Derwent et al., 2013; Derwent, Manning, Simmonds, Spain, et al., 2018).
Recent observational and modelling data have identified a broad $O_3$ maximum in spring and
early summer, aligning with peak stratospheric transport (Ansari et al., 2024; Lin et al., 2012;
Russo et al., 2023). $O_3$ dynamics are complex, and studies reveal discrepancies between model
output and observations (Bessagnet et al., 2016; Vautard et al., 2012), highlighting the need for
further understanding of factors governing $O_3$ levels and trends.



This study investigates the distribution and trends of $O_3$ and its precursors across Ireland,
providing valuable insights into the regional and hemispheric impact on Irish surface $O_3$ levels
and exceedances. By analysing a long term observational dataset, this research highlights
significant seasonal and temporal variations and long-term trends in $O_3$ concentrations.
Advanced modelling results using the Tropospheric Ozone Attribution of Sources with Tagging
1.0 (TOAST 1.0) framework (Butler et al., 2018; Butler et al., 2020) were applied to determine
the drivers of $O_3$ trends in Ireland. Additionally, trajectory analysis is used to trace the origins
of air masses, revealing the impact of transboundary pollution and atmospheric transport. This
integrated approach not only enhances our understanding of the drivers of $O_3$ concentrations,
trends and exceedances over Ireland but also underscores the importance of global and regional
contributions to  $O_3$.
**2. Data and methodology**
**2.1 Observational Network and Analysis Approach**
Measurement data is obtained from the Environmental Protection Agency, Ireland (EPA)
(https://eparesearch.epa.ie/safer/). The $O_3$ monitoring network shown in Figure 1 has been
operational in Ireland since 1994. $O_3$ is measured using an API M400 and $O_3$ analyser based
on UV photometry at all monitoring sites. Measurements of $O_3$ precursors from EPA air quality
monitoring sites are also monitored. The details of measurements site are shown in table 1 .
Numerous previous studies have analysed this data, with a particular focus on the analysis of
Mace Head data to assess background levels of (Carslaw, 2005; Derwent, 1998; Derwent et al.,
1994, 1998, 2001, 2004, 2008, 2013; Derwent, Manning, Simmonds, & Doherty, 2018;
Derwent, Manning, Simmonds, Spain, et al., 2018; Oltmans et al., 2013; Simmonds et al., 2004;
O. P. Tripathi et al., n.d., 2010, 2012, 2013 ). Additionally, the $CH_4$ data is obtained from the



Integrated Carbon Observation System (ICOS) network, accessible at https://www.icos-
cp.eu/data-products/ATM_NRT_CO2_CH4.
For this analysis, the observational sites were classified into three categories: Coastal, Rural,
and Urban, as shown in Figure 1. The classification of the sites is based on Spohn et al., 2022,
with the addition of the coastal category. Hourly data were used to evaluate annual trends based
on monthly mean concentrations. Seasonal analysis   conducted for the four main
meteorological seasons in Ireland, namely Spring (March, April, May), Summer (June, July,
August), Autumn (September, October, November), and Winter (December, January,
February). $O_3$ exceedances were calculated based on the WHO AQGs, indicating that the
maximum daily average over eight hours (MDA8) should not exceed 100 µg/m³. A significant
analysis was performed on data measured at the Mace Head Atmospheric Research Station
(53°33′N, 9°54′ W), which is exposed to pristine marine air masses approximately half of the
time. (Grigas et al., 2017; O'Dowd et al., 2014).
**Table 1.** Details of Environmental Protection Agency Ireland (EPA) $O_3$ measurement sites over
Ireland, with location information, and the data period used for the study.

| Site | Data availability | Type | Latitude | Longitude |
|---|---|---|---|---|
| Mace head | 1994-2022 | Coastal | 53.3253 | -9.9036 |
| Valentia | 2001-2022 | Coastal | 51.9385 | -10.24 |
| Monaghan | 1995-2022 | Rural | 54.0661 | -6.883 |
| Laois | 2005-2022 | Rural | 53.1076 | -7.1983 |
| Kilkenny | 2012-2022 | Rural | 52.6383 | -7.2676 |
| Rathmines | 2002-2022 | Urban | 53.322 | -6.2672 |
| Clonskeagh | 2008-2022 | Urban | 53.3118 | -6.2353 |
| Mayo Castlebar | 2009-2022 | Urban | 53.851 | -9.3003 |
| Swords | 2009-2022 | Urban | 53.4631 | -6.2222 |
| Wicklow Bray | 2009-2022 | Urban | 53.1873 | -6.122 |
| Cork South link road | 2014-2022 | Urban | 51.8785 | -8.4649 |
| Cork Bishops town | 2016-2022 | Urban | 51.8858 | -8.53321 |
| Cork UCC | 2018-2022 | Urban | 51.9 | -8.4863 |






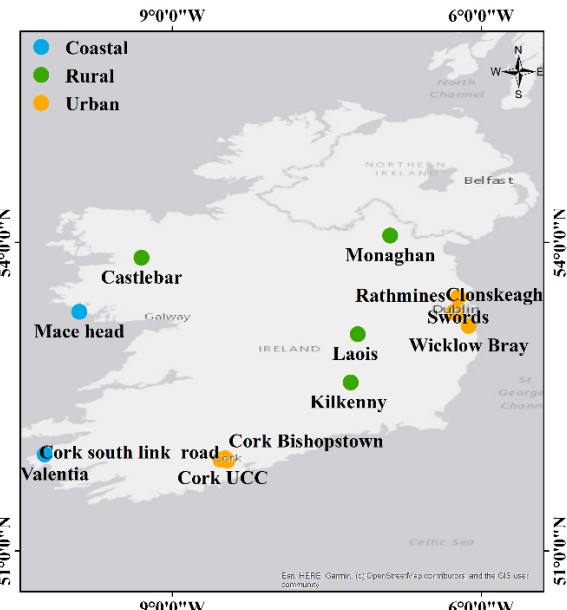


**Figure 1.** – The map of EPA $O_3$ measurement sites over Ireland with classification of backgrounds.

Trend analysis was conducted using the Openair package in R. This software tool is designed for the analysis of atmospheric composition data. Trends were determined using the Theil-Sen slope estimator and Mann-Kendall tests to quantify significance, in accordance with the Tropospheric Ozone Assessment Report (TOAR) guidelines (Lefohn et al., 2018). It is a robust method for estimating trend slopes in time series data, preferable to traditional least-squares regression, which can be sensitive to extreme values and outliers. Uncertainty or reliability of the trend is calibrated according to the p-value, as outlined by (Chang et al., 2023), consistent with the best statistical practices for analysis used in the second phase of TOAR.



**2.2 Clean air sector identification from Back trajectories**


Baseline $O_3$ refers to the concentration of $O_3$ in air masses minimally influenced by local or
regional anthropogenic emissions. Back-trajectory methods are widely used to estimate
baseline $O_3$ levels by analysing the origins and transport pathways of air masses reaching
observation sites. Typically, Lagrangian dispersion models are used to trace air parcels
backwards in time and identify their origin.
For this study, air mass trajectories arriving at Mace Head were calculated using the Hybrid
Single Particle Lagrangian Integrated Trajectory Model (HYSPLIT) (Draxler et al.,2003; Stein
et al., 2015) in conjunction with R software. The air masses were classified into two categories
the clean sector and EU-influenced sector. An air mass was considered part of the clean sector
consider when air mass trajectories remained over the ocean surface for the previous 72 hours.
And the remaining air mass trajectories are classified as the EU-influenced sector.
Meteorological data for the analysis were derived from NOAA reanalysis data (Stunder et al.,
2004). Calculations were performed for 6:00 UTC each day, with a final trajectory height of
100 meters, covering the years 2000 to 2022. The $O_3$ concentrations observed during the clean
sector were averaged to derive baseline levels, consistent with previous studies on baseline $O_3$
trends and sources (Derwent et al., 2013; Oltmans et al., 2006).

**2.3 CAM4-Chem Model**


The CAM-Chem air quality model, part of the Community Earth System Model (CESM),
simulates atmospheric chemistry and the interactions among chemical constituents,
meteorology, and climate. It incorporates detailed chemical mechanisms, emission inventories,
and meteorological data to simulate pollutant dispersion, thereby allowing us to determine air
quality trends. CAM-Chem has been applied in numerous studies, significantly contributing to



the understanding of regional and global atmospheric processes. (Lamarque et al., 2012; Tilmes
et al., 2016). The model features a flexible chemical pre-processor to allow for detailed
handling of atmospheric chemistry. Studies have demonstrated that CAM-Chem accurately
represents conditions in both the troposphere. (Aghedo et al., 2011; Lamarque et al., 2010) and
the stratosphere (Lamarque et al., 2008; Lamarque and Solomon, 2010), including temperature
structure and dynamics (Butchart et al., 2011). Offline CAM-Chem has also been utilised in
the Hemispheric Transport of Air Pollution (HTAP) assessments. (Anenberg et al., 2009; Fiore
et al., 2009; Jonson et al., 2010; Shindell et al., 2008;Tan et al., 2018).
For the current study, we analyse simulations of the Community Atmospheric Model version 4
CAM4-Chem (Community Atmosphere Model version 4 with chemistry) ((Lamarque et al.,
2012). The model simulations were carried out at a horizontal resolution of $1.9° \times 2.5°$,
featuring 56 vertical levels for the 2000-2018 period, with specified dynamics derived from
MERRA2 reanalysis. (Molod et al., 2015). Tagged source attribution of tropospheric
ozone (TOAST 1.0) is a novel tagging methodology developed for the CESM to quantify
source contributions to $O_3$ . Unlike traditional methods that rely on sensitivity simulations,
TOAST uses an online tagging approach to track $O_3$ production from specific $NO_x$ and VOC
sources (e.g., anthropogenic, biogenic, biomass burning, lightning) directly within the model,
allowing for efficient attribution of $O_3$ to regional and sectoral emissions while maintaining
full chemical coupling. The tool has been validated against observations and demonstrates
utility in disentangling the impacts of different emission sectors on $O_3$ pollution. (Butler et al.,
2018, 2020; Lupaşcu et al., 2022; Nalam et al., 2024)
Global CAM4-Chem model simulations are performed for the years 2000-2018 with $NO_x$ and
VOC tagging (as described in Ansari et al., 2025; Nalam et al., 2025), with the base chemical
mechanism (MOZART; Emmons et al., 2012) and source code modified to account for extra
tagged species representing regional and sectoral identities. Anthropogenic emissions of $NO_x$,



CO, and non-methane volatile organic compounds (NMVOCs) are incorporated from the
Hemispheric Transport of Air Pollution version 3 emissions inventory. (HTAPv3; Crippa et
al., 2024), which includes land-based emissions, international shipping emissions, and aircraft
emissions. Biomass burning emissions are sourced from the GFED-v4 inventory (Van Der
Werf et al., 2010), while biogenic NMVOC emissions are derived from CAM4-GLOB-BIO-
v3.0. The $O_3$ source attribution technique used for this study is described in (Butler et al.,
2020)**.**

**3. Results and discussions**
**3.1 Yearly variation of $O_3$**
Figure 2 shows box plots, illustrating the average $O_3$ concentrations for 13 sites over the
duration of the available dataset, as discussed in section 2.1 providing a comprehensive
overview of the variability and distribution of $O_3$ concentration. Coastal sites, such as Mace
Head, show higher $O_3$ levels compared to other sites, with an annual average concentration 77
$\mu g/m^3$. Similarly, Valentia shows higher mean concentrations at 78 $\mu g/m^3$ (in 2003). In urban
areas like Rathmines, Dublin, $O_3$ concentrations remained consistently lower, with averages
ranging from 39 to 56 $\mu g/m^3$. Similarly, South link road and Bishopstown sites in Cork city,
recorded relatively lower concentrations compared to coastal and rural locations, reflecting the
impact of high urban $NO_x$ emissions. Rural sites like Laois and Kilkenny showed intermediate
$O_3$ concentrations, less influenced by urban emissions. These sites consistently show $O_3$
averages ranging between 50 to 57 $\mu g/m^3$, with little variability, highlighting the predominant
role of steady background $O_3$ contributions in rural sites. $O_3$ concentrations vary significantly
with proximity to emission sources – adjacent to urban areas, $O_3$ levels can be lower due to
titration, where $O_3$ reacts with NO, causing $O_3$ depletion, but the transport of precursors can



cause an increase in $O_3$ concentration downwind of the sources  (Jeon et al., 2014; Monks et
al., 2015;Zhu et al., 2012)
The red line over the box shows a clear seasonal pattern in  $O_3$ concentration for each site.
With a spring-time (March-April) peak and summer-time (June-July),  dip, with the highest
peaks in the coastal sites, and lowest dips in urban sites, influenced by local emissions e.g.
Cork South Link Road and Swords.

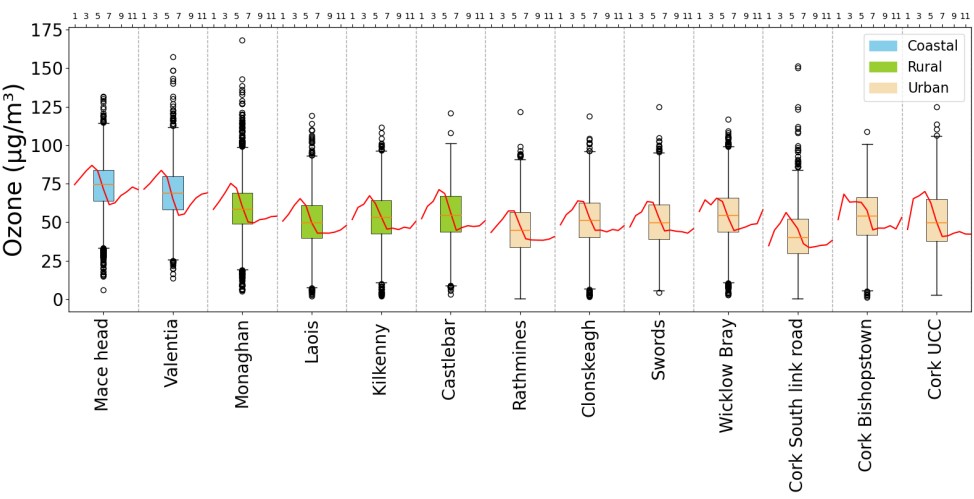


**Figure 2.** Annual average $O_3$ concentration at different sites in Ireland. In each box, the lowest
whisker level represents the 5th percentile, the box spans from the 25th to the 75th percentile,
the horizontal line within the box represents the median 50th percentile, and the upper whisker
represents the 95th percentile.  The average of monthly $O_3$ values calculated for the entire period
of each station, and the red line shows the average monthly $O_3$ variation of all sites top axis
shows the month (1– 12).
**3.2 $O_3$ Trend analysis**
**3.2.1 Yearly trend**





Table 2. summarizes the Theil-Sen trends in $O_3$ concentration (in µg/m³ per year) across 13
monitoring sites in Ireland over different periods: 5 years (2018-2022), 10 years (2013-2022),
15 years (2008-2022), and the available years of data for each site. In the coastal regions, Mace
Head shows a consistent decrease in $O_3$ levels over the 5, 10, and 15-year periods, although the
entire dataset exhibits a small rising trend 0.02 µg/m³ per year. These trends are mostly in
agreement with previous studies, where there was a positive trend observed in background $O_3$
up to the mid-2000s, which stabilised and began to decline in the 2010s (Derwent et al., 2018).
Valentia shows a long-term decreasing trend of -0.23 µg/m³ per year, consistent with the
previous study by Tripathi et al..2010.
In rural areas, Monaghan exhibits a declining trend in $O_3$ concentrations across all time periods,
indicating an overall reduction. Laois shows an upward trend over the 10 and 15-year periods,
though there is a slight decline in the most recent 5 years. Kilkenny presents slight negative
trends over the 5 and 10-year periods (-0.29 and -0.01µg/m³ per year). Negative trends are
observed in Castlebar-(0.71 and -0.05 µg/m³ per year).
The Dublin urban area sites (Rathmines, Clonskeagh, Swords) predominantly show increasing
trends in $O_3$ levels, indicative of changes in urban pollution or local emissions, with decreased
suppression of $O_3$ levels in urban regions due to decreased local emissions. (Derwent et al.,
2024). This is consistent with the "weekend effect," as observed by (Atkinson-Palombo et al.,
2006)whereby a reduction in $NO_x$ due to reduced weekend traffic decreases $O_3$ removal by
$NO_x$ titration, leading to higher surface $O_3$ levels, likely to occur in wintertime, and in regions
with low photochemical production due to low insolation such as Ireland. Mixed results are
observed at the urban stations of Cork. This suggests variable factors affecting Cork $O_3$ levels.
Coastal sites like Mace Head and Valentia generally show decreasing trends, potentially due to
less local emission sources but with more significant impacts from regional and long-range





transport, However, a detailed analysis of the trends requires consideration of seasonal effects.
**Table 2**. - Trends in surface O3 concentration (µg/m³ per year) calculated for 13 sites in Ireland
over different periods over the complete dataset: 5 years (2018-2022), 10 years (2013-2022),
15 years (2008-2022), and the available measurement record for the site. The p-value evaluates
the reliability of the trend, whereas a lower p-value indicates trend certainty. Adopting the trend
reliability scale defined for TOAR-II studies (Chang et al., 2023), trends with very high
certainty will be marked by ***(p ≤ 0.001), trends with high certainty with ** (p ≤ 0.01), and

| Site No | Site name (Classification) | Measurement Record | Trend over record µg/m³ per year | 5-year trend 2018-2022 µg/m³ per year | 10-year trend 2013-2022 µg/m³ per year | 15-year trend 2008-2022 µg/m³ per year |
|---|---|---|---|---|---|---|
| 1 | Mace Head (C) | 1994-2022 | 0.02 | -0.25 | -0.31* | -0.11 |
| 2 | Valentia (C) | 2001-2022 | -0.23*** | -1.15*** | -0.84*** | -0.32** |
| 3 | Monaghan (R) | 1995-2022 | -0.19*** | -0.74** | -0.35* | -0.09* |
| 4 | Laois (R) | 2005-2022 | 0.39*** | -0.15 | 0.3** | 0.46*** |
| 5 | Kilkenny (R) | 2012-2022 | 0.02 | -0.29 | -0.01 | |
| 6 | Castlebar (R) | 2009-2022 | 0.18* | -0.71* | -0.05 | |
| 7 | Rathmines (U) | 2002-2022 | 0.27*** | 1.72*** | 1.15*** | 0.48*** |
| 8 | Clonskeagh (U) | 2008-2022 | 0.33*** | 0.97** | 0.12 | 0.33*** |
| 9 | Swords (U) | 2009-2022 | 0.6*** | 0.07 | 0.33*** | |
| 10 | Wicklow Bray (U) | 2009-2022 | 0.14* | 0.04 | | |
| 11 | Cork South-link Road (U) | 2014-2022 | 0.51* | -0.44 | | |
| 12 | Cork Bishopstown (U) | 2016-2022 | 1.05** | -1.81* | | |
| 13 | Cork UCC (U) | 2018-2022 | -0.94 | -0.94 | | |

low to medium certainty with *(p ≤ 0.05).
**3.2.2 Monthly trend**



Figure 3. shows the monthly trend for 10 years from the period 2012-2022. Mace Head
(coastal) and Monaghan (rural) sites predominantly show a rising trend in winter/early spring
, with a decreasing trend in late spring to summer. Valentia shows a decreasing trend in every
month except February when levels are significantly impacted by long-range transport and
stratospheric sources (Auvray and Bey, 2005; Pan et al., 2018). Urban sites show a general
increasing trend, as yearly trend but with a seasonal signal in Clonskeagh an increase in winter-
spring and a decrease in late spring or summer. Seasonal trends of the 15-year  dataset are
supplied in supplementary figure S2, where coastal stations exhibit a pronounced increase in
early spring and a decrease in late summer, with a consistent near-year-round increase in urban
stations of Rathmines and Laois.

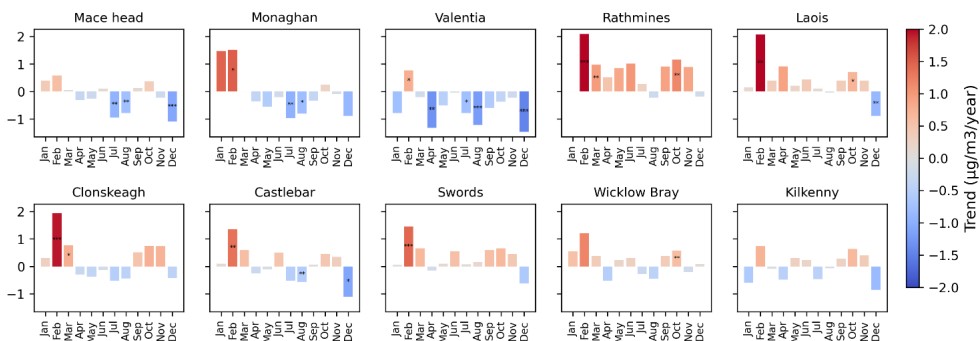


**Figure 3.** Monthly trend analysis of $O_3$ at different sites for 10 year period. (2012-2022)
Adopting the trend reliability scale defined for TOAR-II studies (Chang et al., 2023), trends
with very high certainty are  marked by ***($p \leq 0.001$), , trends with high certainty with **($p$
$\leq 0.01$), and low to medium certainty with **($p \leq 0.05$).  Positive trends are in red shade and
negative trends are in blue shade.




**3.3. $O_3$ Exceedance**

The $O_3$ exceedance are identified according WHO criteria and the results shown in figure 4. It shows the monthly $O_3$ exceedances at 13 sites in Ireland over the available measurement dataset. The highest and lowest numbers of $O_3$ exceedances were observed at Mace Head and Rathmines, representing coastal and urban sites, respectively. Most exceedances occurred in spring when $O_3$ concentrations were at a maximum.

Most sites recorded elevated spring-time occurrence in exceedances. E.g. Rathmines had its highest number of exceedances in April 2019, while Laois reached a peak of 13 exceedances in May 2017. Castlebar and Swords show increased exceedance occurrences in spring and early summer, particularly notable spikes occurring in 2010, 2013, 2016, and 2019. Conversely, Wicklow Bray exhibited a different pattern, showing significant spikes in February and March 2022, alongside occasional exceedances during March, April, and May, for example, in 2012 and 2018. Cork South link Road also recorded exceedances, particularly in March and June, with significant spikes in 2018 and 2019. Cork Bishoptown shows exceedances, especially in February and March 2019, while Cork UCC experienced spikes, particularly in April and May 2019. Kilkenny consistently exhibited exceedances during spring and summer, with April and May often recording the highest number, particularly in 2019. This highlights the impact of seasonal atmospheric conditions on $O_3$ levels. It is noted that summertime exceedances, although less frequent in occurrence, indicate significant photochemical production that would be required to elevate $O_3$ levels from the annual dip in the seasonal cycle to exceed the WHO AQG threshold. These episodic spikes are characteristic of unique climatic or pollution events and warrant further study.



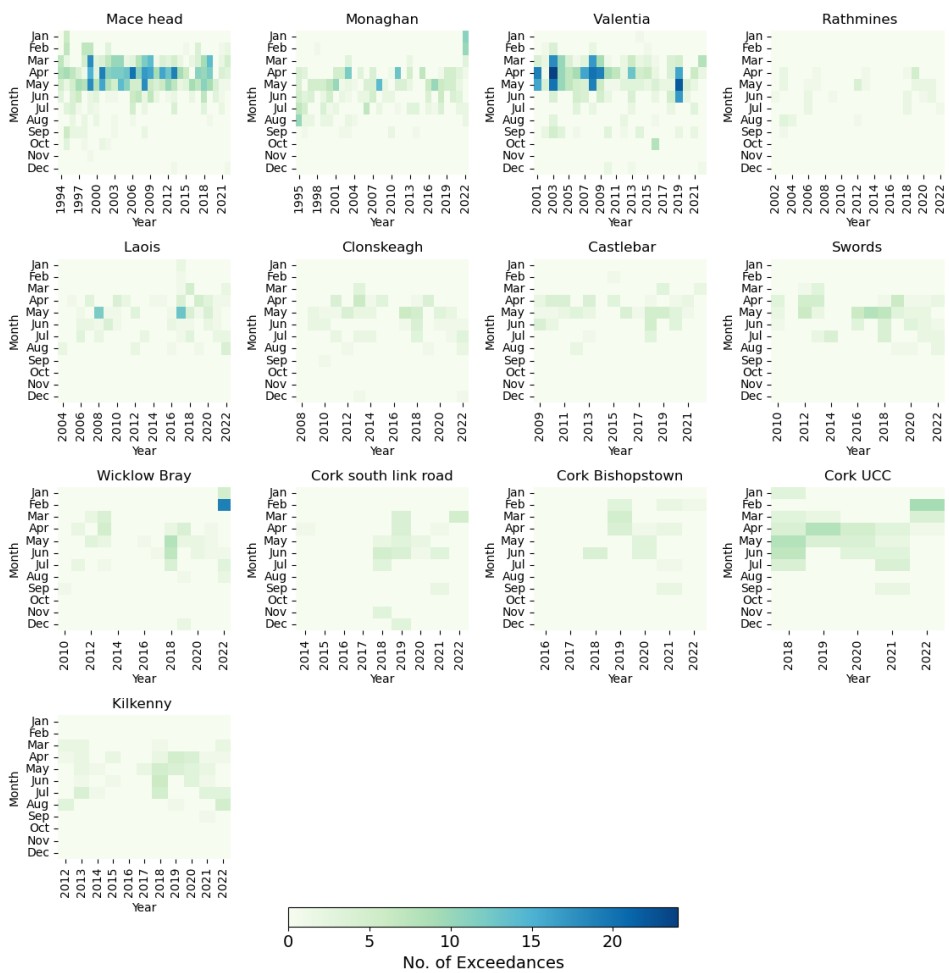

**Figure 4.** - Monthly $O_3$ exceedance at different sites in Ireland.

Figure 5 depicts trends in $NO_2$ and $CH_4$ concentrations across various Irish measurement sites.

Most monitored sites exhibit a decreasing trend in $NO_2$ concentrations because of pollution

control on transportation, industrial activities, and energy production in the EU and North

America, in line with previous studies (Coleman et al., 2013; Donlon et al., 2024).In contrast,

$CH_4$ levels at three sites Mace Head, Malin Head, and Carnsore Point indicate a significant and

persistent rise in $CH_4$ concentrations. Mace Head, known for its clean Atlantic air. Malin Head,

situated at Ireland's northern tip near the UK border, offers a unique position to observe both



clean marine air and transboundary pollution whereas  Carnsore Point in the southeast, is
capture air masses from both the UK and mainland Europe, Carnsore Point receives the
majority of air masses from the land (Spohn et al., 2022). These $NO_2$ and $CH_4$ trends reveal a
dual dynamic: while $NO_2$ levels are decreasing due to effective emission controls, $CH_4$ levels
are rising unabated, highlighting the need for enhanced mitigation strategies targeting $CH_4$.

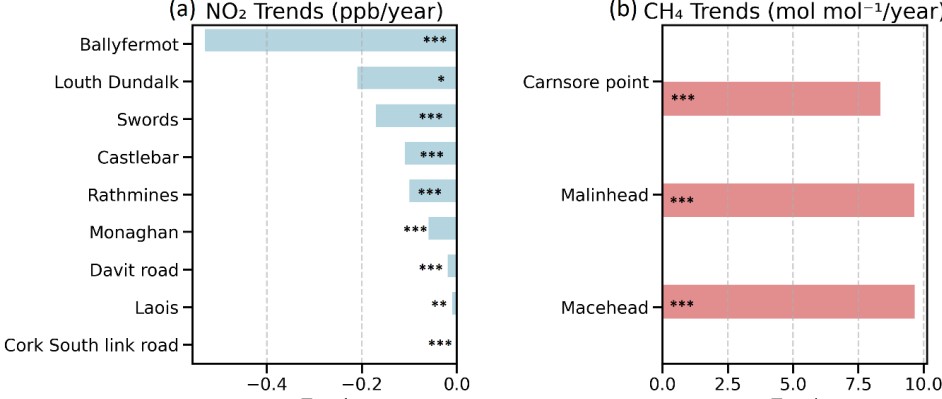


**Figure 5.-** Trend in $O_3$ precursors $NO_2$ (a), and $CH_4$ (b) at different sites. Trends with very high
certainty are  marked by \*\*\*($p \leq 0.001$), , trends with high certainty with \*\*($p \leq 0.01$),, and
low to medium certainty with \*($p \leq 0.05$).
To evaluate the relationship between $NO_x$ and $O_3$ concentrations in an Irish context and the
potential benefit of abrupt enforcement of $NO_x$ control measures, we assess the impact of the
COVID-19 2020 lockdown, Spring 2020, whereby the lockdown period saw a prominent
relative decrease in $NO_2$, yet an increase in surface $O_3$ compared to average measurements for
the same months 2017-2019 in most national monitoring stations (Figure 6). The negative
correlation between $O_3$ and $NO_2$ is indicative of a $NO_x$-saturated regime, normally associated
with polluted urban environments and $NO_x$ titration events. Similar results are discussed by
(Spohn et al., 2022)  with meteorology. This effect was widely observed during the COVID





lockdown (Ordóñez et al., 2020; Tavella & da Silva Júnior, 2021; C. Zhang & Stevenson,
2022). Significant enhancement of $O_3$ occurs at the inland measurement sites, despite a 2020
spring-time decrease in $O_3$ observed at background coastal sites, Mace Head and Valentia.
These coastal stations are less sensitive to changes in European $NO_x$ emissions than inland sites
and more sensitive to stratospheric and hemispheric transport (Tan et al., 2018)
The atmospheric conditions in Ireland do not align with the interpretation of the atmosphere as
being either a $NO_x$-controlled regime for clean environments or a $NO_x$-saturated regime in
polluted environments. The negative correlation between $NO_x$ and $O_3$, due to $NO_x$ titration,
observed in Ireland occurs under relatively clean atmospheric conditions, but it is consistent
with low-insolation conditions, which are characteristic of Irish meteorology and frequent
cloud cover ((Pall E E and Butler, n.d.)

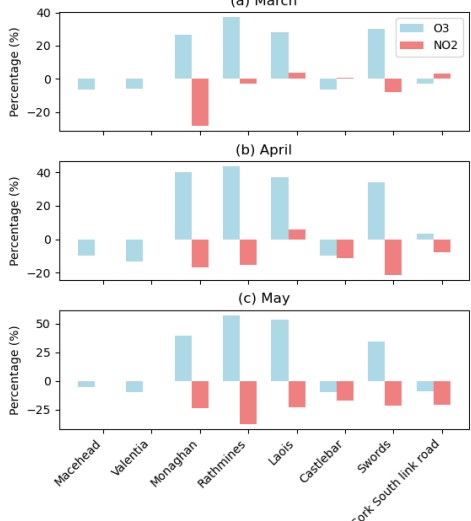



**Figure 6.-** Percentage change in $NO_2$ and $O_3$ during the lockdown period of 2020 as compared
to the 2017-2019 average at different sites in Ireland for (a) March (b) April (c) May Month.
**3.4 Model and Observations Comparison**





### 3.4.1 Comparison between CAM4 – Chem Model and Observations

Global simulations were performed with the CAM4-Chem model enabled with source tagging (Butler et al., 2018) for 2000-2018 and the modelled $O_3$ over Ireland was compared with surface $O_3$ measurements at five sites. Figure 7 shows the comparison of monthly $O_3$ CAM4-Chem and ground station $O_3$ data. From this figure, it is observed that CAM4-Chem exhibits negative (positive) bias in rural and coastal (urban) sites. The underestimation at Mace Head is probably caused by the coarse grid resolution, covering a large area not representative of Mace Head conditions. The influence of coastal meteorology also leads to an underestimation of $O_3$ (Yerramilli et al 2012 ). Coastal meteorology, including cool sea surface temperatures and persistent clouds, suppresses $O_3$ formation. McVeigh et al., 2010 explained this through eddy correlation measurements showing downward ozone fluxes over coastal waters west of Ireland. The dry deposition rate over land would exceed that over the ocean leading to a lower simulated $O_3$ concentration for the entire grid cell. Dry deposition is enhanced by solar radiation (Coleman et al.,2012; Coleman et al., 2013; Pio et al., 2000) hence model measurement discrepancy is at a maximum in late summer months. Overestimation of $O_3$ in Clonskeagh and Cork South link Road is likely due to coarse grid handling of localised emissions and subsequent atmospheric chemistry.



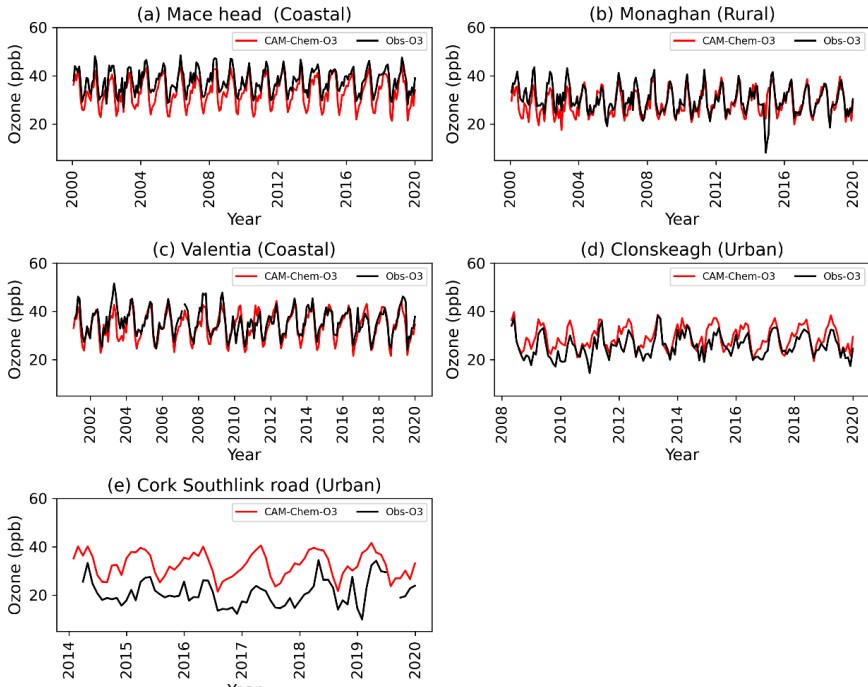

**Figure 7. -** The comparison of Monthly CAM4 – Chem $O_3$ and Monthly $O_3$ observations at five sites in Ireland**.**

At Mace Head, the model shows a negative mean bias of - 4.42 (-11.68% normalized mean bias) but strong correlation (r =0.83). In Monaghan and Valentia, the model shows smaller biases of −1.43 and −1.54, normalized mean biases of −4.74% and −1.54%, and correlation coefficients of 0.73 and 0.72, respectively. These high correlations are in line with (Tilmes et al., 2015). However, at Clonskeagh and Cork South Link Road, the model overestimates, with positive biases (3.38 and 11.98) and weaker correlations (0.68 and 0.49). Statistics are discussed in more detail in the supplementary material table S1. These results suggest better model performance at coastal/rural sites and greater discrepancies in urban areas as expected at this model resolution.

**3.4.2 Source attribution using CAM4-Chem**



To quantify the contribution of various precursor emission sources to modelled $O_3$
concentrations, the TOAST1.0 dual $NO_x$ and VOC tagging technique is utilised as described in
(Butler et al., 2018). This allows us to attribute the modelled $O_3$ to the emissions of $NO_x$ and
VOC precursors across different source sectors and regions as listed in Table 3.  The $NO_x$ and
VOC precursor emissions across different source sectors and regions, as shown in the Table,
are responsible for the attribution of the modelled $O_3$.
**Table 3. -** List of tags used in $NO_x$ and VOC tagging.

| Regional Land-Based Tags | | Regional Oceanic Tags | | Global Sector/Process-Based Tags | |
|---|---|---|---|---|---|
| **ARC** | Arctic | **NAL** | North Atlantic | **AIR** | Aircraft |
| **CAS** | Central Asia | **ENA** | Eastern North Atlantic | **BIO** | Biogenic |
| **EAS** | East Asia | **NAE** | North America East Coast | **BMB** | Biomass Burning |
| **EUR** | Europe | **NAW** | North American West Coast | **LGT** | Lightning |
| **MCA** | Mexico & Central America | **NPA** | North Pacific | **STR** | Stratospheric Intrusion |
| **MDE** | Middle East | **BNS** | Baltic and North Seas | **XTR** | Extra untagged $O_3$ |
| **NAF** | North Africa | **HBY** | Hudson Bay | **CH4** | Methane |
| **NAM** | North America | **IDO** | Indian Ocean | **OCN** | Oceanic Sources (DMS) |
| **RBU** | Russia-Belarus-Ukraine | **MBC** | Mediterranean, Black, and Caspian Seas | **SHP** | Shipping |
| **SAS** | South Asia | **SHO** | Southern hemispheric oceans | **AIR** | Aircraft |
| **SEA** | Southeast Asia | | | **INI** | InitialCondition$O_3$ |
| **VRW** | Rest of the World | | | | |




The monthly tagged major precursor contributions to surface $O_3$ at Mace Head, averaged over
the 2000-2018 simulation period, are shown in Figure 8. The stratospheric source of $O_3$
dominates in Winter-Spring, contributing to the spring-time maxima due to vigorous
stratospheric transport. European $NO_x$ emissions contribution peaks in May, while lightning
$NO_x$ has the greatest impact in winter. North American emissions contribute 3.5 to 5.25 ppb,
peaking in April, and aviation emissions contribute 1 to 3 ppb, with the highest contributions
in winter and spring. Biogenic $NO_x$, significant between June and October, contributes an
average of 3.6 ppb, with higher contributions during August and September. Biogenic VOC
sources contribute slightly more, averaging over 4 ppb during late autumn and maintaining a
more sustained contribution throughout the year. East Asian $NO_x$ emissions, contributing up to
3.6 ppb, show a minimum contribution in July and August. North Atlantic shipping $NO_x$ (NAL)
accounts for up to 2.4 ppb of $O_3$ during July month. The total shipping NOx (SHIP) also
contributes significantly and shows the highest contribution in June month.
Methane ($CH_4$) is the dominant reactive carbon molecule contributing to $O_3$ formation. VOC
emissions from biomass burning also play a measurable role, contributing 1 to 2 ppb, with their
largest contributions in August and September. Finally, European VOC emissions contribute 1
to 3 ppb, with the largest impact from March to May, coinciding with the spring-time peak in
surface $O_3$. These findings allow quantification of specific sources amidst the complex
interplay of regional and global sources in driving seasonal variations in surface $O_3$ levels over
the Irish domain, highlighting the roles of stratospheric processes, anthropogenic emissions,
biogenic sources, and lower-latitude contributions in shaping the observed patterns at
background monitoring sites such as Mace Head.



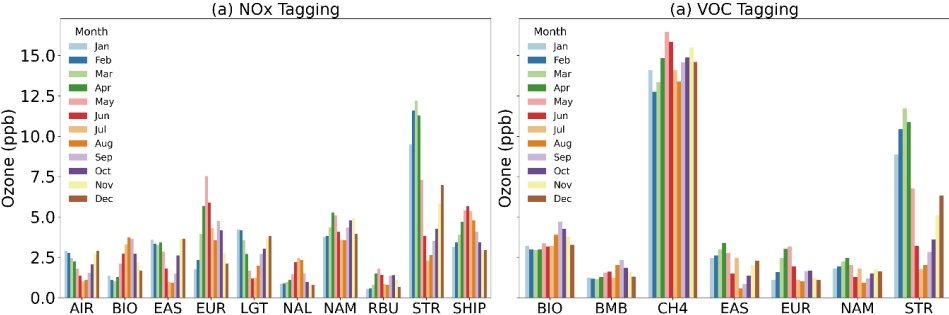

**Figure 8**.- Absolute contribution of major $NO_x$ sources (a) ($NO_x$ Tagging) and VOC source

(b) ($NO_x$ Tagging) to the CAM4-Chem simulated surface $O_3$ for the Mace Head grid cell

between 2000-2018.

Figure 9 shows the monthly changes in contributions to surface $O_3$ at Mace Head over the

simulation period (2000-2018). A negative (blue) trend indicates that the contribution of the

source to simulated surface $O_3$ in this grid cell has declined over the simulation period, whereas

a positive trend (red) indicates the contribution to surface $O_3$ has risen. Figure 9 (a) indicates

that the amount of simulated $O_3$ at Mace Head originating from European or North American

$NO_x$ decreases during the simulation period, consistent with EU & North American emission

reductions over the period (, (Guerreiro et al., 2014;US EPA 2027), with more significant

reduction occurring in late Spring through late summer, when EU $NO_x$ contributions are most

significant to Mace Head $O_3$ concentrations (as seen in Figure 8).

There is a rising trend in simulated surface $O_3$ originating from NOx emissions from global

aviation, from East Asia, and to a lesser extent from South Asia, which is more pronounced in

the wintertime. This seasonality in source contributions explains the observed reduction in

spring-time maxima and increase in winter-time levels from the measurement record. East-

Asian and South-Asian VOCs also contribute to a rising trend in simulated $O_3$, with a more

pronounced increase in winter and spring. This highlights a different pattern in hemispheric $O_3$

contributions, where emission reductions in Europe and North America are accompanied by



increased influence from lower latitudes. This increasing contribution could become a more
important source of background $O_3$ in the future. The contribution of $CH_4$ also has a positive
trend over the simulation period, but the $CH_4$ trend has a reliable correlation only in the
December and spring periods, with no observed trend during the summer months when
atmospheric $CH_4$ trends have very low certainty (correlation coefficient, p>0.33) consistent
with decline in local NOx emissions in Europe.  The Anthropogenic VOC contributions from
Europe (EUR) and North America (NAM) show a negative trend for all months.

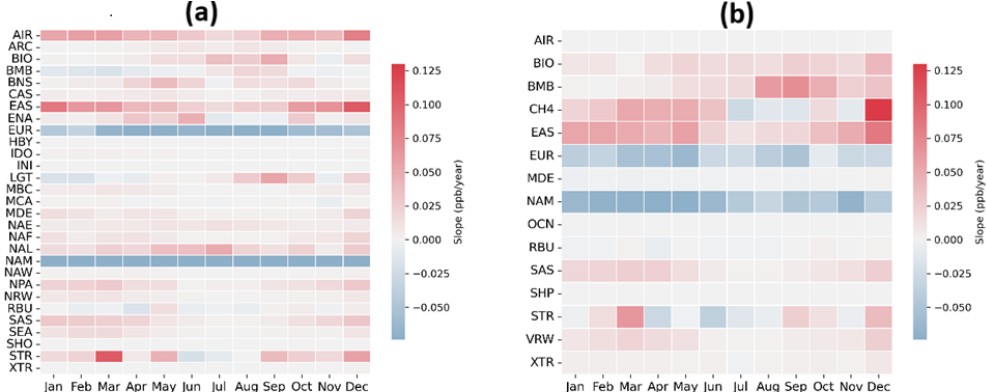

**Figure 9.** Trends in contributions to monthly average modelled Mace Head grid cell surface
$O_3$ at for the 2000-2018 period derived from (a) $NO_x$ tagging and (b)VOC tagging.
Table 4. shows the overall trend in the main contributors to $NO_x$ and VOC tagging. It is
observed that there is an increase in simulated surface $O_3$ originating from $NO_x$ contributions,
from aviation and East Asia, while there is a decrease in European (EUR) and North American
(NAM) $NO_x$ contributions. In VOC tagging, Methane ($CH_4$) and East Asian anthropogenic
VOC (EAS) contribute to a rising trend over the simulation period, whereas anthropogenic
VOC contributions from Europe (EUR) and North America (NAM) show a negative trend.
**Table 4** - Overall Trend in contributions to Mace Head grid cell  $O_3$ simulated by CAM4-Chem
for $NO_x$ tagging and VOC  tagging over the simulation period in units of ppb per year. The





trend with very high certainty is  marked by ***(p ≤ 0.001), , trends with high certainty with
**(p ≤ 0.01), and low to medium certainty with *(p ≤ 0.05).

| NO$_x$ Tagging | | VOC Tagging | |
|---|---|---|---|
| | Slope (ppb/year) | | Slope (ppb/year) |
| AIR | 0.0467*** | CH$_4$ | 0.0590*** |
| EAS | 0.0491*** | EAS | 0.0333*** |
| EUR | -0.0900*** | EUR | -0.0553*** |
| NAM | -0.1243*** | NAM | -0.0670*** |


**3.5 O$_3$ Trends in Background and EU influenced sector  Airmasses at Mace Head**
Although Mace Head is classified as a global background site, quantification of the baseline
pollution levels requires filtering the data to limit the data to that arriving from the clean sector.
Based on the trajectories filtering of  Mace head O$_3$ measurement data discussed in section 2.5.
From figure 10 O$_3$ observations shows clean sector consistently has higher O$_3$ concentrations
than the EU influenced sector especially during the annual spring-time high, indicating a longer
lifetime for O$_3$ over the North Atlantic, and the land mass and pollution sources are acting as a
sink for O$_3$ in the Irish context (Fowler et., al 2008). A decreasing trend in spring-time levels
is observed for both clean and EU influenced sectors, consistent with the decrease in precursor
emissions in Europe and North America. There is a significant difference between clean, and
EU influenced sector measurements, the influence of EU influenced sector  air to scavenge O$_3$
via NO$_x$ titration, leading to higher O$_3$ in clean-sector air masses, as also found in previous
studies (Coleman et al., 2013).An increasing trend is observed in the winter-time EU influenced
sector, which is not observed in the clean sector. This would infer a decrease in winter-time O$_3$
depletion events due to decreasing European emissions (from the EU influenced sector),
consistent with the conclusions from previous studies of Mace Head surface O$_3$ (Derwent et
al., 2024). Summer-time values do not exhibit a notable trend or a discrepancy between clean



and EU influenced sector measurements, indicating that there is little $O_3$ advected into Europe
from the west in the summer months. Autumn values show higher $O_3$ in the clean sector, but
without a significant slope.
The model results indicate that the clean sector consistently exhibits higher $O_3$ concentrations
than the EU-influenced sector, except during the summer season. In the winter season, a
significant increasing trend is observed for both sectors, which aligns well with the $O_3$
observations. A decreasing trend is observed during the summer season, consistent with the
observations for both sectors. In spring and autumn, a positive trend is observed.
Trends in contributors of model $O_3$ during the different seasons for the clean and   EU-
influenced sector are shown in Tables S3 to S4  in the supplementary material. Aviation and
East Asian $NO_x$ shows consistently positive and significant trends in both sectors, while North
America $NO_x$ shows strong negative trends throughout the year in the clean sector.  For the
EU-influenced sector in $NO_x$ tagged (Table S2), similar positive trends observed for aviation
and East Asian, with North America $NO_x$ remaining negative and European $NO_x$ showing more
significant declines in spring and winter.  In case of VOC tagged $O_3$, the East Asian VOCs
shows an increasing trend and North America VOCs negative across all seasons in both sectors
(Table S3 and S4). European VOCs also show a consistent negative trend, particularly strong
in the EU-influenced sector. Methane trends are seasonally positive, especially in spring and
winter.
It is observed that the model consistently simulates $O_3$ at lower concentrations than that
observed at Mace Head. This is not surprising, considering the coarse resolution of the model,
which limits its ability to represent fine-scale processes and dry deposition accurately. Dry
deposition is typically higher over land, and the grid cell covering Mace Head includes land
area, as shown in Figure S5 of the supplementary material. Further, as explained by Fiore et al.



(2009), models average the landscape characteristics within a grid cell, which can enhance $O_3$
deposition and result in lower simulated $O_3$ concentrations; hence, the discrepancy is more
pronounced in the clean sector data.

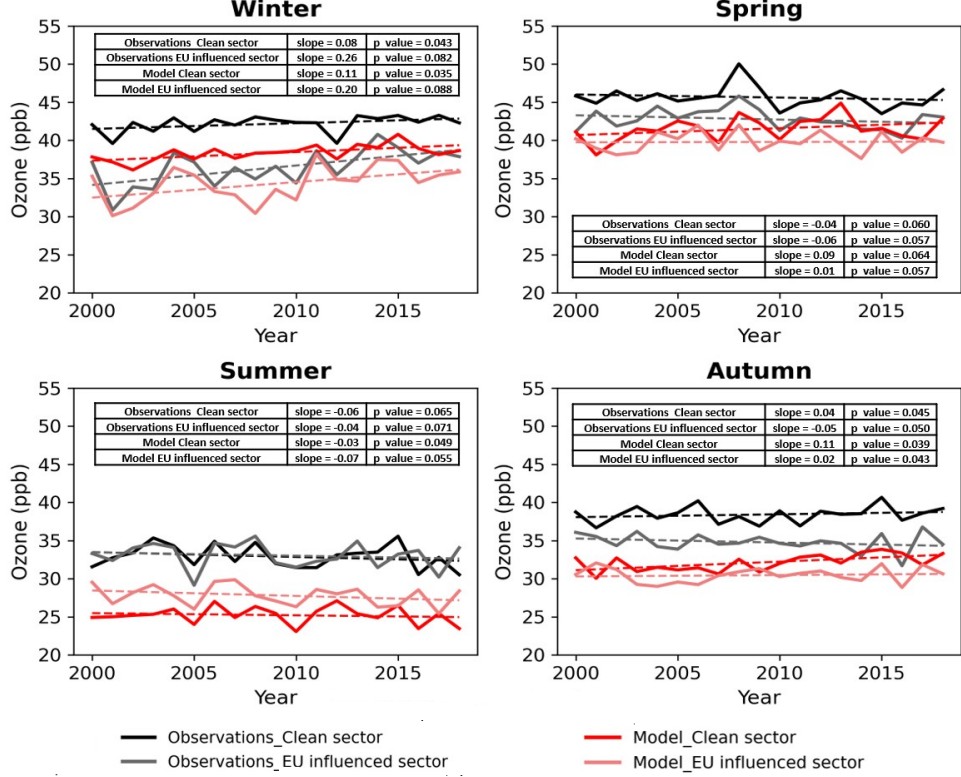


**Figure 10.** - Trend in seasonal Average of observed $O_3$ (black) and Model $O_3$ (red) at Mace

head, separated into clean sector and EU-influenced sector.
**3.6 Exceedances from the clean and EU-influenced sector at Mace Head**
Exceedances observed at Mace Head between 2000 and 2022 are separated into clean and EU-
influenced sectors based on trajectory air masses and shown in Figure 11.  33% of all
exceedances for this period occurred in clean air masses, the remainder occurring when $O_3$-EU



influenced sector air is advected over Ireland and local land masses to enhance surface $O_3$,
which is already elevated at Mace Head compared to inland and urban sites.

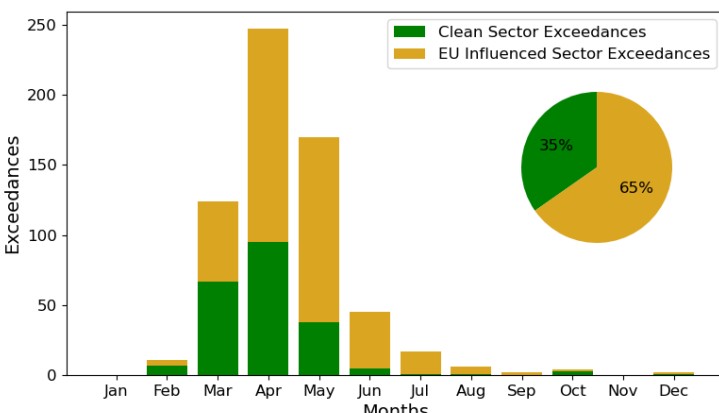

**Figure 11.** - Exceedances measured at Mace Head per month from 2000 until 2022, during the
clean air sector(green) and EU influenced sector (yellow). The percentage of both to total
exceedances is shown in the inlay.
Figure 12  shows the trend in spring-time exceedances and the 95th percentile Spring-time $O_3$
measured at Mace Head between 2000 and 2022. A decreasing trend in exceedances and clean
sector spring-time exceedances is observed, with a greater decreasing trend in the total number
of exceedances. This indicates that the changes that are driving the reduction in the exceedances
in Europe are coming into effect at a quicker rate than the changes that are driving $O_3$ event
reduction over the North Atlantic. The trends in the exceedance counts are not significant,
according to the criteria in Chang et al., 2023, but there is a statistically significant decreasing
trend in the 95th percentile springtime surface $O_3$ over the measurement record. Figure 12 (b)
shows the trend in spring-time surface $O_3$ measured at Mace Head segregated into Clean and
EU-influenced sector.  The trend is more significant both in magnitude and statistical certainty
for the EU-influenced sector, indicating EU emission changes having a more pronounced effect



on spring-time $O_3$ measured at Mace Head $O_3$ as compared to changes affecting $O_3$ transported
or formed over the North Atlantic.

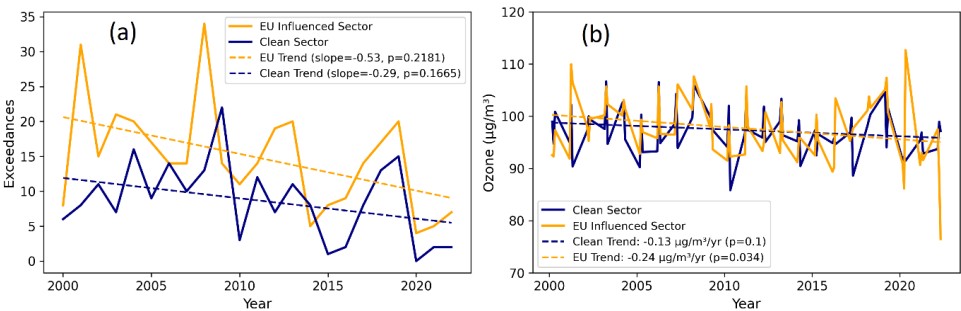


**Figure 12.** (a) The trend in Spring-time exceedances measured at Mace Head between 2000
and 2022 (blue) with the clean-air exceedances (gold), and (b) The trend in 95th percentile of
spring (Mar- May) $O_3$ measured in µg /m$^3$ for the clean sector (blue) and the EU-influenced
sector (gold).
Figure 13 shows monthly cumulative contributions to simulated $O_3$ concentrations within the
Mace Head grid cell for $NO_x$ and VOC tagging during $O_3$ exceedance, which is observed from
$O_3$ observations calculated as discussed in section 3.3. In addition, these exceedances are
categorised into clean and EU-influenced sectors. The maximum exceedances are observed in
March to May month. From Figure 13 (a), it is clear that stratospheric intrusion, North
American $NO_x$, European $NO_x$, and East Asian $NO_x$ are the major contributors driving
exceedances at Mace Head during the spring months (March to May). Among these, European
emissions dominate the supply of $NO_x$ precursors in April, reaching their peak in May. Figure
13 (b) shows that $CH_4$ is the most dominant source, followed by stratospheric intrusion and
Biomass burning. North American and European VOC emissions also contribute significantly
to $O_3$ formation during this period. Collectively, these findings highlight the complex interplay
of regional and global sources in driving surface $O_3$ exceedances over the Irish domain.





The cumulative $O_3$ contributions to EU-influenced sector and clean sector exceedances  for
$NO_x$ tagging and VOC tagging, it is clear that the North American $NO_x$ also contributes
significantly to exceedance in both clean and EU-influenced sectors at Mace Head during
March to May months. It may be due to transport and mixing, regional stagnation or synoptic-
scale recirculation. In the case of VOC tagging, stratospheric intrusion, and $CH_4$ show notable
contributions. Biomass burning, East Asian emissions and North American VOC emissions
also play a role in $O_3$ exceedances.

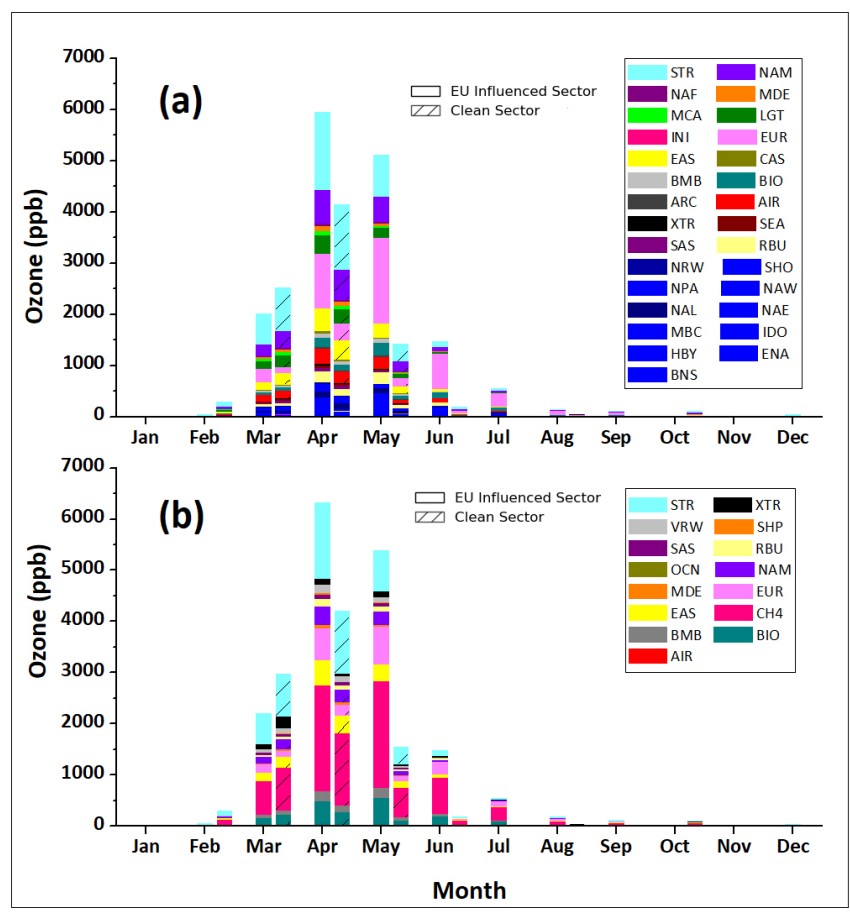




**Figure 13** - Monthly cumulative Mace Head grid cell $O_3$ contributions to EU influenced
sector and clean sector exceedances (a) $NO_x$ tagging and (b) VOC tagging Mace Head grid
cell.

**4. Conclusion**
This study highlights the complexities of $O_3$ pollution in Ireland, revealing that coastal areas
experience higher $O_3$ concentrations than rural and urban environments, attributed to the effect
of transboundary pollution and stratospheric intrusion. Over the last two decades, urban sites
have shown a significant increasing trend in $O_3$ levels, likely influenced by decreasing nitrogen
oxides ($NO_x$) in Europe, including Ireland, and North America. The analysis also points out
that exceedances at coastal monitoring sites correlate with years of higher spring maxima,
driven mainly by hemispheric transport and stratospheric influences(16%). Utilising the
advanced capabilities of the CAM4-Chem model with dual $NO_x$ and VOC tagging, we
identified key factors affecting seasonal $O_3$ variations, such as the spring-time peak and
summer dip, driven by a mix of stratospheric intrusion, hemispheric transport, and regional
emissions. Trend analysis from simulation results identified East Asian and aviation emissions
as significant contributors to the rising winter trends in $O_3$, while reductions in North American
and European emissions accounted for the decrease in spring peaks. This study provides a
comprehensive understanding of the various factors affecting $O_3$ levels in Ireland, offering
important insights for the development of $O_3$ pollution control policies.

**Data availability**
All data are available upon request.



**Author contributions**

LC  designed the study. NK  analyzed the data and wrote the manuscript. TA and TB provided CAM-Chem model results and reviewed the manuscript.. JO and CD reviewed the manuscript and edited it.LC edited it with contributions from all coauthors

**Competing interests**

The authors declare that at least one of the authors sits on the editorial board of ACP.

**Acknowledgement -** The authors acknowledge the Environmental Protection Agency (EPA) of Ireland for their financial support of the Ozone project under the EPA Research Programme 2021-2030 (project number 2022-CE-1133), and the European Union's Horizon Europe Research and Innovation programme under HORIZON-CL5-2022-D1-02 (grant no. 101081430-PARIS).

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






**Figure Captions**

**Figure 1**. The map of EPA $O_3$ measurement sites over Ireland with classification of backgrounds.

**Figure 2**. Annual average $O_3$ concentration at different sites in Ireland. In each box, the lowest whisker level represents the 5th percentile, the box spans from the 25th to the 75th percentile, the horizontal line within the box represents the median 50th percentile, and the upper whisker represents the 95th percentile. The average of monthly $O_3$ values calculated for the entire period of each station, and the red line shows the average monthly $O_3$ variation of all sites top axis shows the month (1– 12).

**Figure 3**. Monthly trend analysis of $O_3$ at different sites for 10 year period. (2012-2022) Adopting the trend reliability scale defined for TOAR-II studies (Chang et al., 2023), trends with very high certainty are marked by ***($p \leq 0.001$), , trends with high certainty with **($p \leq 0.01$), and low to medium certainty with **($p \leq 0.05$). Positive trends are in red shade and negative trends are in blue shade.

**Figure 5**.- Trend in $O_3$ precursors $NO_2$ (a), and CH4 (b) at different sites. Trends with very high certainty are marked by ***($p \leq 0.001$), , trends with high certainty with **($p \leq 0.01$),, and low to medium certainty with *($p \leq 0.05$).

**Figure 6**. Percentage change in $NO_2$ and $O_3$ during the lockdown period of 2020 as compared to the 2017-2019 average at different sites in Ireland for (a) March (b) April (c) May Month.

**Figure 7**. The comparison of Monthly CAM4 – Chem $O_3$ and Monthly $O_3$ observations at five sites in Ireland.



**Figure 8**. Absolute contribution of major NOx sources (a) (NOx Tagging) and VOC source
(b) (NOx Tagging) to the CAM4-Chem simulated surface $O_3$ for the Mace Head grid cell
between 2000-2018.
**Figure 9.** Trends in contributions to monthly average modelled Mace Head grid cell surface
O3 at for the 2000-2018 period derived from (a) $NO_x$ tagging and (b)VOC tagging.
**Figure 10.**    Trend in seasonal Average of observed  O3  (black ) and  Model $O_3$ (red) at Mace
head, separated into clean sector and EU-influenced sector.
**Figure 11**. Exceedances measured at Mace Head per month from 2000 until 2022, during the
clean air sector(green) and EU influenced sector (yellow). The percentage of both to total
exceedances is shown in the inlay.
**Figure 12**.  (a) The trend in Spring-time exceedances measured at Mace Head between 2000
and 2022 (blue) with the clean-air exceedances (gold), and  (b)  The trend in 95th  percentile
of spring (Mar- May) $O_3$ measured in µg /m³ for the clean sector (blue) and the EU-
influenced sector (gold).
**Figure 13**  Monthly cumulative Mace Head grid cell  $O_3$ contributions to  EU influenced
sector and clean sector exceedances  (a) NOx tagging and (b) VOC tagging  Mace Head grid
cell.



