# Peer review of "Surface Ozone Distribution & Trends Over Ireland: Insights from long-1 term measurement record and source attribution modelling 2 3 Nikhil Korhale 1, Tabish Ansari 2, Tim Butler 2, Jurgita Ovadnevaite 1, Colin D.O'Dowd 1, Liz 4 5 Coleman1</"

_EGUsphere, 2025_

## Referee Comment (RC1)

General Comments

This is a nice study of the factors contributing to ozone levels in Ireland. It illustrates the different trends at urban, rural and coastal sites. The addition of the ozone tagging is helpful to understand the contributing factors to these trends. This will be suitable for publication after addressing some points below.

Some of the descriptions around NOx-saturated vs NOx-limited could do with clarifying, particularly with regard to season. It seems reasonable that urban sites will always be NOx-saturated, but it would be surprising if rural sites in Ireland were NOx-saturated outside winter. It would be useful to compare with other studies in similar latitudes (maybe UK?) to see if NOx saturation has been seen in other rural sites. The patterns in figure 3 show rural sites decreasing throughout the spring and summer (except for Laois) which would be consistent with NOx-limited chemistry, but with very strong increases in February which presumably ends up causing a positive annual trend in table 2. Since February ozone rarely causes exceedances, describing these sites as having an increasing trend might imply pollution is getting worse at these sites when it probably isn't getting worse in the peak seasons. Some discussion of why Laois is behaving more similarly to the urban sites would be useful.

Similarly discussion of the contributions of EU and N. American NOx seem to emphasise the importance of NOx titration which is mostly wintertime rather than the increase springtime production.

The discussions in the supplement should be brought into the main paper if they are important, or removed if not.

Specific comments

Line 66-68: This should comment on whether the NAO increases or decreases ozone levels.

Line 82: Is this 40% increase global, EU or Ireland?

Line 183: It might be useful to give the grid resolution in km over Ireland

Section 3 figures: It would be useful to keep the same order of stations throughout the tables and figures, and to group coastal, rural and urban so that a reader doesn't have to repeatedly refer to table 1.

Figure 2: This figure averages over different time periods for different stations. Given the trend in concentrations over time this can distort the comparison. The same period should be used as far as possible.

Line 212: It is not obvious why a single year (2003) has been highlighted for Valentia when all other values are time period means.

Line 277-278: Figure S2 shows coastal increases in February which is conventionally "late winter" rather than "early spring". And then decreases throughout the spring and summer.

Figure 3 needs error bars. I couldn't see the *** markings.

Figure 5: What years are these trends calculated over? Are they the same or different. As with figure 2 the use of different time periods might distort the comparison.

Line 329 to 347: The percentages changes in figure 6 need some uncertainty analysis – are they at all significant? Could much of the change be due to meteorology in 2020 vs that in 2017-2019? Could there be a circulation change that reduced ozone at the coast but increased it inland? Or could changes in insolation enhance ozone production inland but enhance destruction over the ocean? The stronger change at rural sites rather than urban doesn't support NO2 titration being the major contribution to the 2020 increases. The % changes in rural ozone seem very large compared to the NO2 decrease – it would be good to see this in ppb where presumably the concentration change would be even larger. It is not obvious that there is sufficient decrease in NO2 to explain the ozone increase through titration.

 Line 342: This sentence needs to be clearer. Is it referring to urban or rural sites? Average NOx concentrations in each need to be stated and compared with literature studies of NOx-limited vs NOx-saturated conditions.

Line 346: Did Ireland have low-insolation and frequent cloud cover in March, April or May 2020?

Line 356: Figure S5 should be referred to here to illustrate the colocation of the grid and the sites.

Line 428-433: This should discuss that the lifetime of ozone is much longer in winter and hence Ireland can receive transport from further distances. Even in winter insolation and temperatures will still be high enough to produce ozone in South Asia. Discussions of transport should cite HTAP 1 and 2 studies.

Figure 10: It would be useful to see the annual average trends shown too. It looks as if this would be positive for both clean and EU sectors. It looks as if most of the slopes are not significant at the $p < 0.05$ level. This should be commented on.

Line 462-463: It is not evident that there is a net chemical sink for ozone over Ireland in spring since ozone decreases with decreasing emissions. A dry deposition sink seems more plausible.

Line 464: Decreasing spring ozone from the EU influenced sector is not consistent with the increases in rural O3 during COVID. This suggests that the COVID ozone change was not due to emissions.

Lines 466-467: The scavenging of ozone via NOx titration is only true in the winter months. In spring and summer the decrease in ozone with decreasing emissions indicates net chemical production in the EU sector.

Lines 472-474: It is not clear what is meant by "there is little O3 advected into Europe from the west in the summer months". Is this just saying that there is no east-west gradient in ozone therefore eastward or westward advection will have no effect?

Line 481: How are the clean and EU sectors separated in the model data?

Line 505: It should be explained why the exceedances are higher for the EU sector even through the mean concentrations are lower – presumably the variance is higher, which would be interesting to discuss.

Line 515: Should clarify that the greater decreasing trend is for the EU sector.

Line 539: Add "VOC" after "dominant".

Line 543: The first clause in this sentence seems incomplete.

Figure 13: Explain how the quantify plotted is derived. Is it the mean ozone above 100 ug/m3 summed for each day exceeded. The units on the y-axis must include a time dimension, presumably ppb days or maybe ug/m3 days if the exceedance criterion is in ug/m3? I suggest thinking about the colours used, it is difficult to see which shade refers to which region.

Lines 558-560 The increased urban ozone is likely to be dominated by the local NOx changes rather than from Europe and North America as suggested here. The exception might be winter, when ozone is low anyway. This study hasn't demonstrated that European and North American NOx are important in urban areas.

Lines 560-562: This claim of correlation of exceedances at coastal sites with years of higher spring maxima is not supported by any of the text in section 3.3. Note that most of the exceedances come from the EU sector which would suggest an EU rather than hemispheric source.

---

## Author Comment (AC1)

**Response to Reviewer Comments – responses in red text.**

We are extremely grateful to the reviewers for their valuable and constructive comments. We have carefully revised the manuscript and addressed all comments. We believe these suggestions have improved the manuscript.

**Reviewer 1- Comments**

**General Comments**

This is a nice study of the factors contributing to ozone levels in Ireland. It illustrates the different trends at urban, rural and coastal sites. The addition of the ozone tagging is helpful to understand the contributing factors to these trends. This will be suitable for publication after addressing some points below. Some of the descriptions around NOx-saturated vs NOx-limited could do with clarifying, particularly with regard to season. It seems reasonable that urban sites will always be NOx-saturated, but it would be surprising if rural sites in Ireland were NOx-saturated outside winter. It would be useful to compare with other studies in similar latitudes (maybe UK?) to see if NOx saturation has been seen in other rural sites. The patterns in figure 3 show rural sites decreasing throughout the spring and summer (except for Laois) which would be consistent with NOx-limited chemistry, but with very strong increases in February which presumably ends up causing a positive annual trend in table 2. Since February ozone rarely causes exceedances, describing these sites as having an increasing trend might imply pollution is getting worse at these sites when it probably isn't getting worse in the peak seasons. Some discussion of why Laois is behaving more similarly to the urban sites would be useful. Similarly, discussion of the contributions of EU and N. American NOx seem to emphasise the importance of NOx titration which is mostly wintertime rather than the increase springtime production. The discussions in the supplement should be brought into the main paper if they are important or removed if not.

Referrning to the general comment above, we do not wish to infer that there is a nox-saturated photochemical ozone production regime in Ireland. Irish conditions rarely lend themselves to photochemical ozone production, so reduction of NOx leads to decreased O3 removal, as seen in Finch and palmer 2020 - comparable study in the UK. The manuscript has been updated to

reflect this point Similarly, we have revised wording to ensure do not imply that pollution is worse in periods when exceedances are not an issue. We have also commented on the reason Laois is influenced by local emissions.

**Specific comments**

Line 66-68: This should comment on whether the NAO increases or decreases ozone levels.

Response - During a positive NAO phase, surface ozone levels increase. In contrast, during a negative NAO phase (NAO-low), ozone levels decrease. This has been added to the revised manuscript

Line 82: Is this 40% increase global, EU or Ireland?

Response - It is global increase and now mentioned it revised manuscript.

Line 183: It might be useful to give the grid resolution in km over Ireland

Response - Yes, the details of CAM -Chem Model grid in km is added in revised manuscript.

Section 3

figures: It would be useful to keep the same order of stations throughout the tables and figures, and to group coastal, rural and urban so that a reader doesn't have to repeatedly refer to table 1.

Response - Yes, the figures have been revised according to the order in Table 1 and have been added to the revised manuscript

Figure 2: This figure averages over different time periods for different stations. Given the trend in concentrations over time this can distort the comparison. The same period should be used as far as possible.

Response - Yes, the time periods differ for these sites due to the start dates of measurements. The data period is mentioned in in table 1.

Line 212: It is not obvious why a single year (2003) has been highlighted for Valentia when all other values are time period means.

Response - Yes, the values represent time-period means sentence is changed.

Line 277-278: Figure S2 shows coastal increases in February which is conventionally "Late winter" rather than "early spring". And then decreases throughout the spring and Summer.

Response – The sentence in the manuscript has been revised accordingly

Figure 3 needs error bars. I couldn't see the *** markings.

Response - This figure shows monthly trend values, so plotting error bars is not feasible. Instead of using asterisks to show significance, the significant values are highlighted using colour for better visualisation. The updated figure is added in revised manuscript.

Figure 5: What years are these trends calculated over? Are they the same or different. As

with figure 2 the use of different time periods might distort the comparison.

Response - The trends are calculated for the available data period for $NO_2$, the data period varies by site, predominantly covering 2004–2022, with a few sites covers 2012–2022. For $CH_4$, the data cover the period 2010–2022.

Line 329 to 347: The percentages changes in figure 6 need some uncertainty analysis – are they at all significant? Could much of the change be due to meteorology in 2020 vs that in 2017-2019? Could there be a circulation change that reduced ozone at the coast but increased it inland? Or could changes in insolation enhance ozone production inland but enhance destruction over the ocean? The stronger change at rural sites rather than urban doesn't support NO2 titration being the major contribution to the 2020 increases. The % changes in rural ozone seem very large compared to the NO2 decrease – it would be good to see this in ppb where presumably the concentration change would be even larger. It is not obvious that there is sufficient decrease in NO2 to explain the ozone increase through titration.

Response - The strongest changes at Laois. Swords, Rathmines, Monaghan- all inland, and affected by European and surrounding local emissions, whereas the coastal stations are influenced by long-range transport. It is noted that the NOx titration is not the dominant removal mechanism – to look at this, we would need concurrent O3, NO and NO2 measurements at each site, which we don't have and is beyond the scope of this study.

The discussion has been expanded to consider the unique meteorological conditions associated with 2020, with relevant citations and the competing factors governing changes in o3 during lockdown, notably the effect of transport – hence, it is highlighted that it is not the local NO2 changes, but EU wide changes that influenced the increase in O3 during lockdown, . The tagging method faces uncertainties from nonlinear NOx-VOC chemistry, which causes over attribution of local NOx in urban null cycles, (Butler et al., 2018). Emission inventory uncertainties propagate directly to tagged precursor fractions, amplifying errors in source apportionment (Ansari et al., 2025). Untagged stratosphere-troposphere

exchange further biases northern midlatitude surface budgets, limiting the method's accuracy for global ozone attribution (Butler et al., 2018).

Line 342: This sentence needs to be clearer. Is it referring to urban or rural sites?

Average NOx concentrations in each need to be stated and compared with literature

studies of NOx-limited vs NOx-saturated conditions.

Response - Yes, it is the revised in updated manuscript with the references.

The atmospheric conditions in Ireland do not align with the interpretation of the atmosphere as being either a $NO_x$-controlled regime for clean environments or a $NO_x$-saturated regime in polluted environments. The negative correlation between $NO_x$ and $O_3$, due to $NO_x$ titration, observed in Ireland occurs under relatively clean atmospheric conditions, in periods of low-insolation periods and low temperature, which are characteristic of Irish meteorology and frequent cloud cover- conditions that do not facilitate photochemical $O_3$ production (Pall E E and Butler, n.d.) hence $O_3$ levels are influenced predominantly by transport and chemical removal.

Text now reads:

The anti-correlation between $NO_x$ and $O_3$ under relatively clean atmospheric conditions indicate that $O_3$ levels are influenced predominantly by transport and chemical removal, and local photochemical production does not represent a significant surface $O_3$ source owing to periods of low-insolation periods and low temperature, which are characteristic of Irish meteorology and frequent cloud cover (Pallé and Butler, 2002).

Line 346: Did Ireland have low-insolation and frequent cloud cover in March, April or
May 2020?

Response - No ,the explanation is given for the general ozone chemistry over Ireland which is influenced by the insolation and frequent cloud cover.

Line 356: Figure S5 should be referred to here to illustrate the colocation of the grid and the sites.

Response - The reference of Figure S5 has been incorporated into the revised manuscript

Line 428-433: This should discuss that the lifetime of ozone is much longer in winter and hence Ireland can receive transport from further distances. Even in winter insolation and temperatures will still be high enough to produce ozone in South Asia. Discussions of transport should cite HTAP 1 and 2 studies.

Response - The winter conditions here are different from those in South Asia, meaning the temperature does not drop significantly, so ozone production can still occur. The discussion on transport has also been included and cited HTAP1 and HTAP2 studies in the revised manuscript.

Figure 10: It would be useful to see the annual average trends shown too. It looks as if this would be positive for both clean and EU sectors. It looks as if most of the slopes are not significant at the $p < 0.05$ level. This should be commented on.

Response -  The annual average trend is now shown in the supplementary material.

Line 462-463: It is not evident that there is a net chemical sink for ozone over Ireland in spring since ozone decreases with decreasing emissions. A dry deposition sink seems

more plausible.

Response – The dry depositional sink is made explicit in the revised manuscript.

Line 464: Decreasing spring ozone from the EU influenced sector is not consistent with the increases in rural O3 during COVID. This suggests that the COVID ozone change was not due to emissions.

Response - The trends in clean sector and EU influenced sector are calculated for sustained period of decades and the increase in lockdown O3 are observed in the short-term at sites around Ireland, confounded by unique meteorology of 2020. Further, in this section, we focus only on observations at Mace Head, where local sources and hence NOx titration has little influence. The location of the station at Mace Head is made explicit in the discussion.

Lines 466-467: The scavenging of ozone via NOx titration is only true in the winter months. In spring and summer, the decrease in ozone with decreasing emissions indicates net chemical production in the EU sector.

Response –

Text now reads:

The figure shows that the clean sector has consistently higher $O_3$ concentrations than the EU influenced sector for Winter, Spring and Autumn, with most significant disparity between clean and EU sectors in winter/spring when stratospheric intrusion and lightening $NO_x$ contribute most significantly to $O_3$, as discussed in Section 3.4.2.with $O_3$ originating from EU airmasses susceptible to dry deposition and removal via local pollution while traversing the land-mass towards Mace Head, leading to higher $O_3$ in clean-sector air masses consistent with previous studies (Coleman et al., 2013).

Lines 472-474: It is not clear what is meant by "there is little O3 advected into Europe

from the west in the summer months". Is this just saying that there is no east-west gradient in ozone therefore eastward or westward advection will have no effect.

Response – The intention was to state that the air flow from the west is not a dominant source in the summer months, but the sentence has been removed for clarity.

Line 481: How are the clean and EU sectors separated in the model data?

Response - For the model the clean and EU sectors separated based on the same method use for the trajectory analysis the same period taken as clean and EU sectors.

Line 505: It should be explained why the exceedances are higher for the EU sector even through the mean concentrations are lower – presumably the variance is higher, which would be interesting to discuss.

Response -

Exceedances observed at Mace Head between 2000 and 2022 are separated into clean and EU influenced sectors based on trajectory air masses and shown in Figure 11. 35% of all exceedances for this period occurred in clean air masses, the remainder occurring when $O_3$-EU influenced sector air is advected over Ireland and local land masses to enhance surface $O_3$, which is already elevated at Mace Head compared to inland and urban sites.

It is notable that there is a higher proportion of exceedances that occur from EU- influenced sector, despite higher mean observations from the clean sector for all seasons, bar summer. This occurs because of an enhancement of surface $O_3$ occur during an influx of polluted air from EU, UK or local sources. The EU influence on exceedance becomes more proportionally prominent in late Spring and summer, with more frequent easterly airflow and there is a higher occurrence of stagnation events.

Line 515: Should clarify that the greater decreasing trend is for the EU sector.

Response – this is addressed accordingly

Line 539: Add "VOC" after "dominant".

Response - Added in revised manuscript.

Line 543: The first clause in this sentence seems incomplete.

Response - It is revised in the revised manuscript.

Figure 13: Explain how the quantify plotted is derived. Is it the mean ozone above 100 ug/m3 summed for each day exceeded. The units on the y-axis must include a time dimension, presumably ppb days or maybe ug/m3 days if the exceedance criterion is in ug/m3? I suggest thinking about the colours used, it is difficult to see which shade refers to which region.

Response -   First, the exceedances were identified, and then these exceedances were divided into two sectors the EU-influenced sector and the clean sector. Figure 13 presents the hourly ozone exceedance cumulative concentrations in ppb, along with the contributions from different parameters. It indicates which parameters contribute more to the exceedances, in both EU-influenced sector or the clean sector.  In this the ug/m3  to ppb conversion is considered for the exceedances. This explanation is added in revised manuscript.

Lines 558-560 The increased urban ozone is likely to be dominated by the local NOx changes rather than from Europe and North America as suggested here. The exception might be winter, when ozone is low anyway. This study hasn't demonstrated that European and North American NOx are important in urban areas.

Response – the manuscript has been adapted to explicitly acknowledge the role of local NOx, and we have amended the text to allude to the reduction in winter time ozone depletion events due to regional transport of European outflow of CO & CFCS (Derwent et al 2024, Simmonds and Derwent 1991)

Lines 560-562: This claim of correlation of exceedances at coastal sites with years of higher spring maxima is not supported by any of the text in section 3.3. Note that most of the exceedances come from the EU sector which would suggest an EU rather than hemispheric source

Response – Thanks, this was badly articulated – the majority of exceedances occur in the spring months, coinciding with spring maximum which peaks in April. As discussed above, the spring maximum influenced by stratospheric transport, hemispheric and LRT, and EU pollutions pushes over the exceedance threshold, but this was determined from the simulation results, so this sentence has been revised.

**Reviewer 2 - Comments**

This paper addresses an important topic: surface ozone ($O_3$) trends across Ireland, using long-term observational data and model (CAM4-Chem + tagging via TOAST 1.0) source-attribution analyses. The combination of measurements, trajectory / sector classification, and modelling is commendable. The regional focus on Ireland particularly the background coastal site at Mace Head adds value, since many $O_3$ studies focus on larger continental areas. The methods are reasonably described, and the results are of interest for air-quality / atmospheric chemistry / policy communities. That said, the manuscript requires substantial improvement in several areas before it is ready for publication: grammar & sentence structure need polishing, some

methodological choices need clarification, some results could be better contextualized, and a number of references appear missing or inconsistent.

Response - We sincerely thank the reviewer for their valuable and constructive comments. We have carefully revised the manuscript to address the concerns raised regarding grammar and sentence structure and all references have been checked and updated for accuracy and consistency.

**Major concerns**

1. The abstract and introduction present the study aims, but the phrasing is sometimes confusing. For example: "Using innovative trajectory analysis, $O_3$ concentrations, exceedances and were identified by sectors…" (L17–19) – the sentence is awkward, and a verb seems missing ("and … were identified").

   Response - This sentence is rephrased in revised manuscript and much of the abstract has been rewritten to improve flow

2. It would help to explicitly state the hypotheses (e.g., "We hypothesise that urban sites will show increasing $O_3$ due to reduced $NO_x$ titration while background coastal sites will show declining $O_3$ thanks to precursor reductions"). A clear statement of hypotheses will strengthen the framing.

   Response –Many thanks for the suggestion. The authors suggest that the study aims to identify trends, patterns and drivers of changes, rather than testing a hypothesis. In this context, the article is more investigative in nature. However, we have included a description of the analysis in terms of surface ozone sources at the end of the introduction

3.  The authors mention "innovative trajectory analysis" but more precisely explain what is new compared to previous work. Many previous studies have done back-trajectory classification. Clarify what is novel.

    Response – In this work, we used 22-year, 72-hour back trajectories, which were filtered into "Clean sector " and "EU Influenced sector " air masses. This approach is particularly important due to the unique geographic location of Mace Head, situated on the west coast of Europe and in close proximity to the Atlantic Ocean. Understanding the origin and transport mechanisms of clean air masses is crucial in this context. These clean air pathways establish an essential baseline. By characterizing them, our method goes beyond tracking pollution, it quantifies the exposure to natural background conditions and reveals how large-scale circulation patterns influence ozone levels. This allows for a more precise attribution of ozone exceedance events, distinguishing between changes driven by transport from the North Atlantic Ocean and from Europe. It builds upon previous non-hierarchical clustering mechanisms by differentiating between trajectories that have passed over land, and those uninfluenced by land-based emissions - allowing exclusion of continental and local emission sources on ozone concentrations instead of clustering via air-mass origin. Therefore, we mentioned it as this innovative methodology, but more accurate to describe the analysis as advanced trajectory analysis as opposed to innovative. We adopt this more accurate terminology in the revised manuscript.

4.  I suggest a thorough pass with a native English speaker or professional editing service to improve readability, grammar, and logical flow.

Response - We would like to thank the reviewer for the constructive suggestions to improve to the paper's readability and structure. The paper has been revised by authors for readability.

5. The trajectory classification: The criteria (72h over ocean) for "clean sector" should be justified more clearly. Why 72 h? Why 100 m height at 6:00 UTC? Are results sensitive to these thresholds?

Response - The 72h duration captures regional/long-range transport without trajectory error from meteorological uncertainties. The 100m arrival height mixed-layer air sampled by surface monitors, representing boundary-layer flow. The 06:00 UTC aligns with synoptic times and can match daily ozone cycles or measurement periods. This is added in the revised manuscript.

6. The modelling: The CAM4-Chem grid resolution is coarse ($1.9° \times 2.5°$) (L229). This is acknowledged by the authors in discussing biases (L486–490). However, the implications for interpretation (especially for urban sites) should be more emphasised. Is the coarse resolution adequate for urban site comparisons?

Response – Agreed that CAM-Chem has a coarse spatial resolution, it is still appropriate for this study because it provides a reliable representation of the regional background atmosphere influencing the urban site. The model captures large-scale transport patterns, seasonal variability, and background ozone levels, all of which are essential for interpreting urban observations. By comparing urban measurements with regional-scale CAM-Chem outputs, we can distinguish local pollution effects from broader atmospheric processes. The purpose of using CAM-Chem here is not to reproduce fine-scale urban hotspots, but to understand the larger chemical environment surrounding the city. The text is updated to reflect this.

7. The source tagging via TOAST is described, but I would recommend including a validation of the tagging method (or refer to validation studies) in a little more detail. For instance, what is the error/uncertainty associated with the tagging?

Response - The references for the validation studies of the TOAST tagging method, including information on errors and uncertainties, have been added in the revised manuscript.

8. Statistical trend methods: The authors use Theil-Sen slopes (L180–185). It may help to compare with alternative methods or at least to discuss the limitations (e.g., non-stationarity, autocorrelation).

Response -

In this trans analysis theil-Sen method was chosen to quantify the trends magnitude. Mann-Kendall primarily provides a p-value, while Theil-Sen directly provides robust slope estimate. For this study it is required a reliable measure of change over time for direct interpretation and comparison, which aligns with Theil-Sen method. This provides the trend's actual impact, such as the rate of increase per year. This limitations of Mann-Kendall tests are added in revised manuscript.

9. In modelling vs observations comparison (Section 3.4.1), only five sites are used. Why these? Are they representative? Could more sites be used or reasons given for the selection?

Response - The CAM-Chem model covers the 4–5 grids region over Ireland. These 5 sites placed in different domains, representing different backgrounds and having longterm measurements therefore these 5 sites are best suited for comparison of observations and model.

10. The result that urban sites show increasing $O_3$ trends (e.g., Rathmines +0.27 µg m$^{-3}$ yr$^{-1}$ for full period) (L363–365) is interesting. But the mechanism is briefly mentioned ("weekend effect", $NO_x$ titration) (L311–313); this could be developed further, perhaps linking to local emission inventories or changes in VOC/$NO_x$ ratios over time in Ireland.

Response - This is expanded on in the manuscript, and it is emphasised that the effect is related to the role of NOx in the removal of ozone, noting that Irish conditions rarely allow photochemical production of ozone due to frequent cloud cover and low insolation. This effect is supported by referring to UK study (Finch and Palmer, 2020) which observed similar surface ozone increase coinciding with NOx reduction.

11. The discussion of the 2020 COVID-19 lockdown (Section 3.3) is interesting (L451–458). But the data are limited (March-May 2020) and a more nuanced discussion of meteorology confounding effects would strengthen the claim.

Response -

The manuscript has been updated to discuss the confounding effect of meteorology during lockdown- see below.

"It is noted that April and May 2020 had unique meteorological conditions compared to previous years, with lower windspeed, less rain and significantly higher solar radiation, see Figure 12 in Spohn et al. (2022). These meteorological conditions would potentially facilitate photochemical O3 production, contributing to positive O3 anomalies during lockdown period in addition to NOx reduction, also potentially enhancing dry deposition to the ocean. Further investigation into this topic would warrant model sensitivity studies, beyond the scope of this current work."

12. The authors report a decline in spring-time O$_3$ at Mace Head but an increase in winter trends (L465–470). This is a key finding. However, the discussion linking these to emission reductions, stratospheric intrusion, hemispheric transport is somewhat speculative and could be better supported by citations or sensitivity tests.

Response: This section has been rewritten in the revised manuscript, and removed speculation, instead supporting the key findings with the TOAST model results, which serve thepurpose of the sensitivity test – attributing the simulated ozone to the tagged sources rather than repeating perturbed simulations. Decrease in spring-time at Mace Head is clearly due to fairer weather and EU emissions when photochemical production is yet very limited, so NOx controlled. Increase during winter could only be due to (1) photochemistry – not happening in the Northern Hemisphere during winter, (2) decrease in NOx – not the case in winter either OR (3) stratospheric entrainment due to stormier weather in winter and (4) less ozone destruction due to lack of radiation during winter. (3) and (4) are the only plausible reasons for increase. Over the continents, EU, NA, any increase in ozone due to lack of radiation is negated by higher continental NOx emissions, plus stratospheric entrainment is less likely due to more stratified atmosphere over the continents.

13. It would help to compare the Irish trends with trends elsewhere (Western Europe, North Atlantic) more thoroughly — e.g., are the trends consistent with broader European background ozone literature?

Response - The comparison of this trend with other are discussed in revised manuscript to show consistency with most relevant previous studies in European context, citing Yan et al 2018 and Nelson and Drysdale 2025. The North Atlantic context is supported by references to studies of Derwent.

14. The uncertainties in modelling and measurement (especially for attribution of sources) should be more explicitly discussed.

Response - The detail discussion added in revised manuscript in methodology section for both the measurements, the tagged modelling and the theil-sein slope estimation for trend analysis

For modelling,

The tagging method introduces uncertainties from nonlinear NOx-VOC chemistry, which causes over attribution of local NOx in urban null cycles, (Butler et al., 2018).Emission inventory uncertainties propagate directly to tagged precursor fractions, amplifying errors in source apportionment (Ansari et al., 2025). Untagged stratosphere-troposphere exchange further biases northern midlatitude surface budgets, limiting the method's accuracy for global ozone attribution (Butler et al., 2018).

**Minor Comments:**

1. Line 229 - It would be useful to maintain the same order of stations throughout all tables and figures.

   Response - The order of stations in all figure and table is maintained in revised manuscript.

2. Line 326 - Figure 5 Why these NO2 and CH4 sites are selected.

Response - These sites were selected because long-term NO₂ and CH₄ data are available for them

3. Line 342 - This sentence needs to be rewritten for clarity; the current wording is difficult to follow.

Response - Sentence is rewritten in revised manuscript.

4. Line 353 Include details of the model grid used over Ireland to support the interpretation of spatial results.

Response - Details of the model grid used over Ireland now it is added to the revised manuscript.

Line 415 Figure 8 Clarify what the SHIP parameter represents. It is not defined in Table 3.

Response -The SHIP parameter is used as the addition of all oceanic emissions now it is added to the revised manuscript

5. Line 456 - Please provide details of the method used to define background and EU-influenced airmasses. This information is essential.

Response - The detailed methodology for define background and EU- influenced airmasses is added in revised manuscript.

6. Line 543 Check this sentence; it appears incomplete and needs revision.

Response - The sentence is revised in manuscript.

7. Line 551 Figure 13: Change the colour scheme to improve readability.

Response - The colour scheme is changed in revised manuscript

8. Supplementary material - Section 3 is missing. Please correct the numbering and ensure figures.

Response -The supplementary material placed in corrected order in revised manuscript.